# *Tonantzin*, a New Genus of Bess Beetle (Coleoptera, Passalidae) from a Montane Subtropical Forest in Central Mexico, with a Review of the Taxonomic Significance of the Mesofrontal Structure in Proculini

**DOI:** 10.3390/insects10070188

**Published:** 2019-06-28

**Authors:** Cristian Fernando Beza-Beza, Larry Jiménez-Ferbans, Dave J. Clarke, Pedro Reyes-Castillo, Duane D. McKenna

**Affiliations:** 1Department of Biological Sciences, Center for Biodiversity Research, University of Memphis, Memphis, TN 38152, USA; 2Facultad de Ciencias Básicas, Universidad del Magdalena, Santa Marta 470004, Colombia; 3Instituto Nacional de Ecología, Xalapa, Veracruz, Mexico

**Keywords:** Passalidae, New Genus, Cofre de Perote, *Tonantzin*, *Tonantzin tepetl*, Mesofrontal Structure

## Abstract

Mexico has the third highest diversity of passalid beetles in the World. Here we describe *Tonantzin*
**new genus**, a new monotypic genus, potentially endemic to the mountains of central Mexico. The new genus is diagnosed by a new configuration of characters from the mesofrontal structure (MFS) in addition to other characters. The MFS in Passalidae has been treated either as a composite complex character or a combination of individual characters. Using a broad taxonomic sample within Proculini, we discuss the taxonomic and systematic implications of the MFS for the tribe. We define the MFS type *tepetl*. Given the importance of the MFS for passalid taxonomy we propose a new delimitation of the structure using boundaries based on internal and external head structures. We argue that the treatment of the MFS as a complex character better captures the nature of this structure but we ultimately find a need to standardize the way in which this structure is described in the taxonomic literature and used in phylogenetic analyses.

## 1. Introduction

The family Passalidae, commonly known as bess beetles, is a group of subsocial saproxylophagous beetles found mainly in forested habitats and containing approximately 900 described extant species [1]. Most bess beetles live in the tropics (they collectively have a pantropical distribution) and in Mexico most of the diversity of Passalidae is found in the subtropical region. However, a small number of species are known from the Nearctic Region. Mexico has the third highest species richness of bess beetles after Colombia and Brazil, with 93 described extant species [2,3]. These species all belong to the tribes Passalini or Proculini in the subfamily Passalinae. One of the 17 genera of Proculini known from Mexico (*Yumtaax* Boucher, 2006), and 57% of known Mexican species of Proculini are endemic (known only from Mexico), and most of this endemism is associated with, or restricted to, high-elevation montane areas [1,4]. Consequently, these beetles have long been of interest to evolutionary biologists and biogeographers. 

The generic concepts for New World Passalidae are largely those of Reyes-Castillo [5], who divided species among the tribes Proculini and Passalini and provided the delimitation of each genus. In Proculini, he mainly based the generic concepts on the combination of character states derived from key cephalic characters, the most important of these being (a) the shape of the anterior border of the clypeus, (b) the presence of the frontoclypeal suture, (c) the position of the internal tubercles, and (d) the type of mesofrontal structure (MFS) (Figure 1a). The species-level taxonomy of Proculini has remained relatively stable since 1970. Although changes to some generic concepts were proposed by Boucher [1], not all of those changes have been followed by other passalid workers [6,7]. Taxonomic work on Proculini in Mexico has mainly focused on the description of new taxa, either in isolation, e.g., [6,7], or in the context of larger group revisions, e.g., [8,9]. The last genus to be described for Mexico was *Yumtaax*, species of which represented a new circumscription of the *Petrejoides* “*recticornis*” species group *sensu* Castillo & Reyes-Castillo [8], as well as the recognition of a new rank.

In this paper, we describe a new genus from Mexico so far known only from one locality in the Cofre de Perote, Veracruz. Although the taxonomy of Proculini has been relatively stable, the boundaries for certain genera are still unclear (e.g., *Petrejoides* Kuwert, 1896, *Popilius* Kaup, 1861, and *Odontotaenius* Kuwert, 1896). In addition to the generic description, in which the status of *Tonantzin* (new genus) as a distinct lineage of Proculini is abundantly clear on the basis of morphology, we also performed a molecular phylogenetic analysis to independently test the placement of *Tonantzin tepetl* n. sp. within Proculini. The MFS of *Tonantzin* n. gen. is clearly divergent from that of any of the four standard types described by Reyes-Castillo [4]. Therefore, we reconsider the taxonomic value of the four standard MFS types as the main basis for classifying genera of Proculini as proposed by Reyes-Castillo [4,5], and discuss the validity of these MFS types as a taxonomic character in Proculini (and to a lesser extent, Passalini). 

### Taxonomic Significance of the Mesofrontal Structure

In Passalidae classification, the head capsule (and especially the fronto-clypeus) is considered to provide the most informative morphological variation for genus- and species-level taxonomy [1,5]. The mesofrontal structure (MFS) is usually the most prominent structure on the frons (Figure 1) and historically has been treated either as a collection of distinct and presumed independent characters or as a single complex character. For example, Kaup [10] divided the MFS into only two characters, the cephalic horn and the transverse tubercles. More recently, however, Boucher [1] broadened the morphological concept or definition of the MFS to include several distinct characters, including the central tubercle (equivalent to the cephalic horn of Kaup), latero-posterior tubercles (equivalent to the transverse tubercles of Kaup), tentorial tubercles, latero-post-frontal areas, and the occipital sulcus. The presence or absence of the individual characters that comprise the MFS *sensu* Boucher [1] was important in his species delimitations. Alternatively, Reyes-Castillo [5] also defined the MFS as a complex or composite structure and considered it to comprise the central tubercle (equivalent to the cephalic horn of Kaup) and two transverse carinae and lateral tubercles (equivalent to the transverse tubercles of Kaup). For Proculini, he delimited four different “types” of mesofrontal structures that he named the *striatopunctatus*, *bicornis*, *marginatus* and *falsus* types (Figure 1b–e; these discussed in detail in the Discussion). 

Following the work of Reyes-Castillo [5], the MFS types have been used as a key taxonomic character in Proculini for the concepts of genera and for species delimitation. The genus *Spurius* Kaup, 1971, for example, can be diagnosed by having the *bicornis* type, among other characters, and the genus *Popilius* was distinguished from other genera of Proculini (e.g., *Petrejoides*) by having the *marginatus* type [5]. All species of *Oileus* Kaup, 1869 are likewise characterized by the *striatopunctatus* type [5,11], among other characters. The majority of proculine species and genera have the *marginatus* type, with the *falsus* type being the next most common. The *striatopunctatus* type is characteristic of species of *Odontotaenius*, *Oileus* and *Pseudoarrox* Reyes-Castillo, 1970, and occurs rarely in species of other genera (e.g., *Yumtaax cameliae* Beza-Beza et al., 2017). 

Sixteen of the 20 genera proposed by Reyes-Castillo [5] have only one type of MFS. Although other genera do not exclusively have one type of MFS the general trend is that the majority of species within a genus share a specific MFS type. In some cases, a few species within a genus may have a different type from the predominant one. For example, the species *Proculejus ganglbaueri* Kuwert, 1896 and *Proculejus pubicostis* Bates, 1886 are characterized by the *falsus* type, whereas the remaining 5 *Proculejus* Kaup, 1868 species have the *marginatus* type [5]. The inclusion of *P. ganglbaueri* and *P. pubicostis* within *Proculejus* is well supported based on other diagnostic characters of the genus (e.g., position of the internal tubercles close to and sometimes interrupting the fronto-clypeal suture, and the presence of pubescence on the lateral interstriae). 

Since the work of Reyes-Castillo [5], 38 publications (Appendix A) have described new genera or species of Proculini. Of these, 21 papers have explicitly used a specific MFS type (as described above) in the species description, and others have included an MFS type as a diagnostic generic character. However, after the work of Boucher [1], it has become more common to treat the MFS with a stronger emphasis on individual parts of the structure instead of as a complex character (i.e., as MFS “types”).

## 2. Materials and Methods

### 2.1. Terminology, Specimen Preparation and Examination

All descriptions use the morphological terminology of Boucher [1] for dorsal characters of the head and Reyes-Castillo [5] for the rest of the body unless otherwise indicated. Material additional to that of *Tonantzin* and used for the phylogenetic analyses and comparative morphology is listed in Appendix B. Identifications were done by comparing specimens with museum material and by using the following keys: Reyes-Castillo [12,13], Schuster & Cano [14], Pardo-Lorcano [15,16], Reyes-Castillo and Jiménez-Ferbans [17]. When we provide data from specimen labels the “|” symbol represents line breaks and “//” indicates label breaks. 

For comparative morphology, we examined pinned, cleared and alcohol-preserved specimens with a Leica MZ12.5 dissecting microscope. Head capsules of exemplar specimens representing different MFS structure types were disarticulated from the prothorax and cleaned by soaking overnight in a solution of distilled water and detergent, then briefly ultrasonically cleaned to dislodge remaining debris. Cleaned specimens were transferred to 70% ethanol, then 95% and 100% ethanol prior to air drying and mounting on aluminum stubs with carbon adhesive tabs (Electron Microscopy Sciences, Hatfield, PA, USA), and then further dried in a vacuum desiccator before coating with gold-palladium (5 nm) using an EMS 550X sputter coater. Scanning electron micrographs of the MFS and surrounding structures were taken on a Field-Emission Scanning Electron Microscope (Nova Nano SEM 650 from FEI/Thermo Fisher Scientific, Hillsboro, OR, USA) (Integrated Microscopy Center, University of Memphis). 

To gain preliminary insight into the homology of individual MFS structures (*sensu* Reyes-Castillo, [5]), we examined the relation of the MFS to internal and surrounding sclerotized structures in a broad selection of cleared, bleached and stained specimens. The same species used for SEM imaging were studied in this way, plus specimens of other exemplar species representing the diversity of MFS types in Passalinae (Proculini, Passalini) and Aulacocyclinae (see Appendix B). Following the general clearing and bleaching methods described by [18], head capsules from specimens preserved in 100% ethanol were disarticulated from the prothorax and cleared in 10% KOH heated in a water bath on a hotplate. When cooled, they were transferred to distilled water, left overnight, and transferred back to 70% ethanol overnight. These specimens were then bleached and cleaned in a weak (3%) solution of hydrogen peroxide with a few drops of ammonium hydroxide, ultimately removing most pigment and rendering them partly transparent. The bleached heads were transferred to 70% ethanol, stained with Chlorazol Black to better visualize membranous structures and the tentorium, and finally transferred into glycerin for observation. Mouthpart structures were subsequently disarticulated from the head and remaining undigested internal tissue removed with fine forceps.

Focus-stacked habitus and other images were taken with a Canon EOS 70D DSLR camera (Canon, Tokyo, Japan) mounted on a StackShot rail and operated via the image stacking program Zerene Stacker 1.04 (zerenesystems.com/cms/home). Resulting images were edited and adjusted in Adobe Photoshop and Illustrator. Specimens used in the comparative morphology study have been given Passalidae MesoFrontal Structure index numbers of the form “PMFSxx”.

Material Examined for this paper will be deposited in the Colección Entomólogica del Instituto de Ecología (A.C. Xalapa, Mexico; IEXA), the Field Museum of Natural History (Chicago, IL, USA; FMNH); the personal collection of A. R. Gillogly (ARGC); the personal collection of Cristian Fernando Beza-Beza (CFBB), and the Duane D. McKenna voucher collection (DDMC). Acronyms in quotations are not necessarily official collection acronyms. All DNA specimens used were stored in 100% ethanol at −20 °C. Specimen vouchers used for DNA extraction are deposited in DDMC, except for the paratype and allotype of *Tonantzin tepetl*.

### 2.2. DNA Extraction and PCR Protocols 

Extraction of genomic DNA was done using the OmniPrep kit from G Biosciences (St. Louis, MO, USA). All amplifications were done using the Qiagen (Hilden, Germany) PCR core Kit in a Thermo Hybaid PxE 0.2 thermal cycler. For the amplification of CAD, a semi-nested amplification approach was used, as described by Wild and Madison [19]. An initial PCR reaction was performed using the 439f/688r primer set, followed by a secondary amplification of the previous PCR product with the 439f/668r primer set. The PCR reactions for CAD had a final volume of 20 μL each including 1× CoralLoad PCR buffer, 0.2 μM dNTP mix, 2.5 μM MgCl2, 0.75 μM of each primer, and 1.0 μL of DNA (concentration was variable for each sample). For the 439f/668r fragment, the volume of DNA was modified to 0.1 μL of 439f/688r PCR product. The following cycling pattern was used for the amplification of CAD: (1) 94 °C for 2 min, (2) 94 °C for 30 s, (3) 50 °C for 1 min, (4) 72 °C for 1 min, steps 2–4 were repeated for 36 cycles, followed by (5) 72 °C for 5 min. The ribosomal subunit 28s was amplified using the Rd1.2a/Rd4.2b [20] and Squirtle [21]/Rd5b [20] primer combinations. The PCR reactions for 28s had a final volume of 21 μl each including 1.20× CoralLoad PCR buffer, 1.20× Q solution, 0.11 μM dNTP mix, 1.79 μM MgCl2, 0.71 μM of each primer, and 1.0 μL of DNA (concentration was variable for each sample). Both fragments were amplified following a modified cycling pattern from Beza-Beza et al. [9]: (1) 94 °C for 2 min, (2) 94 °C for 40 s, (3) 52 °C for 40 s, (4) 68 °C for 2.5 min, steps 2 to 4 were repeated for 30 cycles, followed by (5) 68 °C for 5 min. Primer sequences are described in Appendix A.

The PCR products were run in a 1.5× agarose gel, extracted, and purified using QIAGEN QIAquick (Hilden, Germany) columns gel extraction kit. The final eluted PCR products were sequenced at Eurofin Genomics LLC (Louisville, KY, USA) using standard Sanger sequencing methods. Sequences were assembled in Geneious 11.1.5 (https://www.geneious.com) and exported to files in FASTA format. Sequences for CAD were aligned in MEGA X [22] using the Clustal W algorithm, followed by a secondary alignment with the Muscle algorithm. 28s sequences were aligned in MAFFT version 7 [23] using the Q-INS-I iterative refinement method to account for the secondary structure of RNA sequences.

### 2.3. Phylogenetic Analyses 

We performed a phylogenetic analysis of partial sequences from the nuclear protein-coding (NPC) gene CAD and the ribosomal DNA subunit 28s. For GenBank accession numbers see Table A1 in Appendix C. The ingroup was comprised of 20 species of Proculini and Passalini. To test the placement on *Tonantzin* n. gen. within Proculini three species of Passalini (the tentative sister group to Proculini; Boucher [1]) were included: *Passalus* (*Pertinax*) *convexus* Dalman, 1817, *Passalus* (*Passalus*) *interruptus* (L., 1758), and *Paxillus leachi* MacLeay, 1819. In addition to *Tonantzin tepetl* n. sp., 16 other species of Proculini were included: *Oileus sargi* (Kaup, 1871), *Oileus rimator* (Truqui, 1857), *Ogyges tzutuhili* Schuster and Reyes-Castillo, 1990, *Ogyges laevissimu* (Kaup, 1868), *Odontotaenius striatopunctatus* (Percheron, 1835), *Odontotaenius disjunctus* (Illiger, 1800), *Petrejoides tenuis* Kuwert, 1897, *Yumtaax recticornis* (Burmeister, 1847), *Yumtaax jimenezi* Beza-Beza et al., 2017, *Verres cavicollis* Bates, 1886, *Verres furcilabris* (Eschscholtz, 1829), *Veturius oberthuri* (Hincks, 1933), *Vetrius assimilis* (Weber, 1801), *Pseudacanthus aztecus* (Truqui, 1857), *Chondrocephalus purulensis* (Bates, 1886), *Chondrocephalus granulifrons* (Bates, 1886). *Leptaulax dentatus* (F., 1792) was included as an outgroup; this species is a representative of Leptaulacini, one of the three Old World tribes of Passalinae. These three tribes are hypothesized to form a clade sister to the New World tribes [1].

Phylogenetic trees were inferred by parsimony, maximum likelihood (ML) and Bayesian analyses of the combined data set comprising both genes. Parsimony analysis was performed in PAUP version 4.0a [24]. A heuristic search for the most parsimonious trees was conducted with 100 random additions. A bootstrap analysis with 1000 replicates, each involving a heuristic search, was done to calculate statistical support for nodes. Partitions for the ML and Bayesian analyses were established using PartitionFinder V.1.1.1 [25]. The PartitionFinder analysis suggested three partitions, one including the first and second codon positions of CAD, a second comprised the 3rd codon position of CAD, and a third partition comprised 28s. The best-fit substitution models for each partition were K80+G, K80+I+G and GTR+I+G, respectively. The ML analyses were performed in IQTree version 1.6.7.2 [26]. Bayesian analyses were implemented using MrBayes version 3.2 [27], (four independent runs of 10,000,000 generations and trees sampled every 1000 generations, with the first 20% of trees discarded as burn-in).

## 3. Results

### 3.1. Phylogenetic Analyses

The final sequence alignment consisted of 23 terminals with 2260 sites; 615 characters were variable of which 385 were parsimony informative. The ML, parsimony and Bayesian analyses recovered the same tree topology (Figure 2) and contained two main clades corresponding to the two tribes of Passalidae known from the New World: Passalini and Proculini. Specimens of *Tonantzin tepetl* n. sp. were recovered within the tribe Proculini, as expected based on morphology. All proculine genera with more than one representative species were monophyletic. *Tonantzin tepetl* n. sp. was the sister taxon of *Pseudacanthus aztecus* and did not render any genus paraphyletic. Thus, our analyses support the taxonomic conclusion that *Tonantzin* n. gen. is an independent lineage within Proculini. 

### 3.2. Taxonomy

#### 3.2.1. *Tonantzin* Beza-Beza, Jiménez-Ferbans, & Clarke n. gen. 

**Zoo Bank:**http://zoobank.org/urn:lsid:zoobank.org:act:61F21FF8-763A-40F4-ABE6-05ACD3C65882.

**Type species:***Tonantzin tepetl* n. sp., by present designation.

**Etymology:** The name *Tonantzin* was the term used in Mexica mythology to refer to female deities. As per article 30.2.2 of the International Commission on Zoological Nomenclature (IZCN), the gender of the genus name *Tonantzin* is feminine.

**Species included:***Tonantzin* n. gen. is a monotypic genus including only *Tonantzin tepetl* n. sp., described below. 

**Description:** Head (Figure 3b–d). Labrum with anterior border concave and anterior angles rounded. Frontoclypeus narrow, dorsally exposed. Mediofrontal tubercles present. Frontoclypeal suture present, but weakly defined. Frontal area and frontal fossae glabrous, smooth, impunctate. Posterior frontal ridges arched, well defined, starting at posterior bases of MFS and joining supra-ocular ridges. Mesofrontal structure of the *tepetl* type, with apex of the central tubercle free, without transverse carina, and with latero-posterior tubercles large and expanded posteriorly (Figure 4). Supra-orbital ridge bituberculate, bifurcated on posterior half. Postocular pits indistinguishable. Occipital sulcus marked and complete. Mentum with central area protruding, lateral lobes fully pubescent with basal fossae weak (shallow). Hypostomal process widely separated from mentum. Antennal club with three short lamellae, terminal antennomere with apex rounded. Maxilla with lacinia bidentate. Ligula tridentate; central tooth slightly larger than lateral teeth. Middle labial palpomere 1.6 x wider than distal palpomere, subequal in length to distal palpomere. 

Mandibles. Apex bidentate; teeth equal in size. Internal inferior tooth on right mandible monocolumnar; on left mandible bifid, without small basal tubercle. 

Thorax. Anterior angles of pronotum rounded; marginal sulcus very narrow, with fine punctures; lateral fossae very weak. Prosternellum rhomboidal, elongated (Figure 3e). Mesosternum scars small, glabrous and opaque. Mesepimeres glabrous. Metasternum with marginal fossae narrow, narrower than mesotibia. 

Elytra. Humeri and epipleura glabrous. Anterior vertical face scarcely pubescent. Striae well defined, with fine rounded punctures more defined in lateral striae. 

Legs. Profemur: longitudinal anteroventral groove present. Protibia: apex of spur acute, down-curved. Mesotibia: longitudinal dorsal ridge well defined, long and heavily pubescent. Meso- and metatibia without spines laterally.

#### 3.2.2. *Tonantzin tepetl* Beza-Beza, Jiménez-Ferbans, & Clarke n. sp. 

**Zoo Bank:**http://zoobank.org/urn:lsid:zoobank.org:act:80A66C26-F625-4B83-95F9-4D330C4E9E54.

**Type material.** Three specimens (two male, one female). **Holotype:** male. Label data, [hand written]: “MEXICO. Veracruz.| Cofre de Perote.| 19°31′39.4″ N 97°4′43.5″ W| 2870 m. 10-septiembre-2016 | P. Reyes-Castillo, E. Arriaza & C. Suárez cols.” // “Punto3|145” // “Genero nuevo”; [printed] “MEXICO: Veracruz: Cofre de Perote,| N 19°31′39.4″ W 97°4′43.5″| 2870 msnm. 10.ix.2016| P. Reyes-Castillo, E. Arriaza,| C. Suárez (cols.) // “*Tonantzin tepetl* Beza-Beza et al. | 
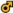
, HOLOTYPE| desig. Beza-Beza et al. 2019.” (IEXA). **Paratypes:** female. Label data, [hand written]: “? Gen. nov.| det. C. F. Beza-Beza 2013”; [printed] "MEXICO: Veracruz: Municipio Xico,| Ingenio El Rosario, 2920 msnm.| N 19°30′57.3″ W 97°5′32.2″,| 16.vii.2012 P. Reyes-Castillo, C.| Beza-Beza, O. Villerías (cols.)” // “McKenna DNA voucher| DDM3175” // “B197S9” // “*Tonantzin tepetl* Beza-Beza et al.| 

, ALLOTYPE| desig. Beza-Beza et al. 2009.” Designated as Allotype (FMNH). Male. Label data, [hand written]: “? Gen. nove | det. C. F. Beza-Beza 2011”; [printed] "MEXICO: Veracruz: Municipio Xico,| Ingenio El Rosario, 2920 msnm.| N 19°30′57.3″ W 97°5′32.2″,| 16.vii.2012 P. Reyes-Castillo, C.| Beza-Beza, O. Villerías (cols.)” // “McKenna DNA voucher| DDM2883” // “B196S16” // “*Tonantzin tepetl* Beza-Beza et al.| 
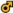
, PARATYPE| desig. Beza-Beza et al. 2009.” (FMNH).

**Etymology:** The name *tepetl* is a noun in apposition and is the Nahuatl (an indigenous Mexican language) term for mountain. In combination with *Tonantzin*, the species name means “Our Goddess of the mountains”.

**Description:** Habitus (Figure 3a). Length 28 mm. Body black, shiny. Head (Figure 3b–d). Labrum with lateral border straight, pubescent; pubescence absent behind concavity. Frontoclypeus oblique, anterior border straight to slightly concave with central notch; latero-frontal tubercles rounded, situated below and smaller than mediofrontal tubercles; mediofrontal tubercles large, acute and pointing forward and outward. Frontoclypeal suture only faintly indicated by row of punctures, erased in middle. Internal tubercles absent. Frontal area and frontal fossae glabrous, smooth and impunctate. Mesofrontal structure type *tepetl*; central tubercle thick with very free apex surpassing anterior cephalic border (Figure 4); postfrontal ridges united with supra-orbital ridges; latero-posterior tubercles large and elongated (Figure 3c,d). Supra-orbital ridges with both tubercles rounded and subequal. Anterior cephalic angles outwardly projecting and rounded. Ocular canthus with rounded apex; in dorsal view protruding slightly to lateral border of eye. Eyes large; ocular canthus 1/3 eye length in lateral view. Hypostomal process glabrous, separated from mentum by more than half its width, reaching anterior border of mediobasal area of mentum. Mentum with basal fossae oval, pubescent and shiny; mediobasal area protruding and with row of setiferous punctures along posterior margin. Antennal lamellae short (h longer than w; Figure 3b).

Mandibles. Width of dorsal mandibular tooth in lateral view 1/3 width of mandible. 

Thorax. Pronotum with scarce punctures restricted to marginal sulcus; anterior angles rounded; anterior border sinuate; lateral fossae impunctate, glabrous. Prosternelum elongate, with truncate posterior apex (Figure 3e). Mesosternum scarcely setose, setae covered by prosternelum; lateral area opaque and scars weakly marked; posterior corner of mesepisternum and mesepimere glabrous. Scutellum with abundant punctures. Metasternum pubescence limited to 3–5 setae in anterolateral area around mesocoxal cavity; disk not delimited by punctures; lateral fossae glabrous and narrow, narrower than epipleura; posterior border of metasternum glabrous. 

Elytra. Shiny, anterior border rectangular. 

Wings. Fully developed.

Legs. Anterior ventral border of profemur with weak groove (Figure 2e) reaching apical pubescence but erased in its proximal half; protibia with dorsal groove complete.

Abdomen. Visible abdominal tergites glabrous; sternite VI with marginal groove distinct, narrow and complete.

Aedeagus. Basal piece fused to parameres in ventral view (Figure 3f–h). Median lobe with *ca*. 50% of ventral surface sclerotized, length 1.5× length of basal piece and parameres together, measured at ventral midline.

**Variation:** In the paratypes there are abundant punctures on the posterior area of the mentum. These are scarce in the holotype.

**Distribution**: This species is known only from a single locality in Cofre de Perote, Veracruz, Mexico.

#### 3.2.3. Taxonomic and Systematic Remarks

Originally this species was misclassified as *Petrejoides jalapensis* Kuwert, 1986 (specimens of this species are housed in the personal collection of Pedro Reyes-Castillo, in IEXA). The misclassification could be due to the overall size, length of the central horn, presence of a frontoclypeal suture, and the geographic distribution of both species. But besides these characters, *Tonantzin tepetl* is clearly morphologically distinct from *P. jalapensis*. For instance, *Tonantzin* does not agree with the current concept of *Petrejoides*, a genus with unclear circumscription [1], and one that has also been recovered as polyphyletic [9]. *Petrejoides sensu* Boucher [1] is circumscribed based on a single autapomorphy––the convex, curved and narrow anterior border of the clypeofrons. For *Tonantzin tepetl*, although narrow, the anterior border of the clypeofrons is straight to slightly concave. In addition to the aforementioned morphological differences, our phylogenetic analyses clearly recover *Tonantzin* as phylogenetically distant from *P. tenuis*, the type species of *Petrejoides* (Figure 2). *Tonantzin tepetl* shares more morphological characters with *Veturius* Kaup, 1871 and *Arrox* Zang, 1905 than with *Petrejoides*. For example, these three genera share a sinuate anterior pronotal border and short antennal club lamellae. In the phylogenetic reconstruction presented by Boucher [1], *Arrox*, *Veturius,* and *Verres* Kaup, 1871 formed a clade. 

The phylogeny shown in Figure 2 supports the relationship between *Verres* and *Veturius* as suggested by Boucher [1], but does not support a close relationship between these two genera and *Tonantzin*. Morphologically, *Tonantzin* differs from *Veturius* in both the presence of a frontoclypeal suture (absent in all known species of *Veturius*) and absence of the latero-frontal fossae (“Aires latéro-frontales” [1]). Additionally, the presence of punctures in the marginal sulcus of the pronotum distinguishes *Veturius* from *Tonantzin*; this character is shared with *Arrox* and other genera of Proculini. *Tonantzin* can also be distinguished from *Arrox* based on male genital morphology, in which the dorsal sclerotization of the phalobase is not extended in *Tonantzin* but extended in *Arrox*. Furthermore, *Tonantzin* has the anterior border of the labrum strongly concave, like the condition in *Verres*; however, it lacks the excavation behind the concavity (a diagnostic character of *Verres*) and *Tonantzin* has a distinct frontoclypeal suture. In our phylogenetic reconstruction, *Tonantzin* is recovered as sister to *Pseudacanthus*, from which *Tonantzin* is distinct. Among the differences between these two genera is the form of the frontoclypeal suture; in *Pseudacanthus* it is strongly indicated and interrupted by the internal tubercles whereas in *Tonantzin* it is only faintly indicated, and the internal tubercles are absent.

## 4. Discussion

### 4.1. The Mesofrontal Structure and the Systematics of Proculini

The mesofrontal structure [5]; (“mediofrontal structure”/”medio-post-frontal structure” of some recent workers) has long been important in the classification of Proculini and Passalidae in general, e.g., [8,12,28]. As demonstrated by recent taxonomic and phylogenetic studies, e.g., [9,29,30,31], it is clear that characters of the mesofrontal structure (MFS) and their states continue to form important components of the generic- and species-level classification of Proculini. However, we highlight a distinct shift in the application of the MFS to Proculini taxonomy, recognition of which prompted us to explore the morphology of the MFS in light of the discovery of *Tonantzin*. Prior to Boucher [1], MFS “types” were in widespread use for taxonomy in Proculini (Appendix A), but literature appearing after this publication seems to suggest that passalid workers have largely abandoned the use of Reyes-Castillo’s [5] MFS types. Instead, recent studies are treating distinct features of the MFS as individual characters in both taxonomic and phylogenetic analyses, e.g., [1,30,31,32]. This shift in usage raises the question of whether there is any value in considering MFS types as a systematic character and, moreover, it questions how the MFS type we identify in *Tonantzin* is fundamentally different from the other four types. Furthermore, Boucher’s [1] concept of the MFS is much broader than that of Reyes-Castillo’s [5], incorporating several other characters that are questionably related to the MFS. Therefore, we suggest that there is a need to morphologically circumscribe this structure and to define the boundaries and homology of component structures using explicit criteria that can be applied across taxa. Although a rigorous attempt to solve this problem is beyond the scope of the present paper, herein we provide some observations based on cleared specimens that can form the basis for a future and more elaborate comparative study aiming to provide a homology-based circumscription of the MFS. Such a limitation would recognize the tightly integrated nature of this complex structure, which we feel justifies the continued use of the MFS “type” system in Proculini and perhaps more broadly in Passalidae.

The four MFS types defined by Reyes-Castillo [5] are largely based on the presence or absence of three main externally visible structures: the central tubercle (usually the major character), lateral tubercles or transverse carinae (equivalent to the latero-posterior tubercles of Boucher [1]), and frontal ridges (equivalent to posterior frontal ridges of Boucher [1]). The presence and development of other tubercles and minor carinae and their connections with or between these more prominent structures are also included as components of the MFS. In the *striatopunctatus* type, the central tubercle is free, directed upwards from the base and bent forwards almost 90°. Lateral tubercles are usually absent or when present are poorly defined, but the transverse carinae are always absent (Figure 1b). The *bicornis* type is fundamentally different from the *striatopunctatus* type and all other types, being characterized by the absence of the central tubercle and the presence only of the lateral tubercles. These can be (but are not always) joined by a carina (Figure 1c). The third and fourth types of mesofrontal structures seemingly differ in only minor ways. In the *marginatus* type the apex of the central tubercle is fused with a single or pair of longitudinal carinae that extend anterior to it and join with the frontal ridges; the transverse carinae are always present and extend to the side of the base of the central tubercle (Figure 1d). The lateral tubercles are seemingly absent or poorly developed. In the *falsus* type the apex of the central tubercle is directed forwards or upwards and the frontal ridges, when present, meet the central tubercle at the base, not the apex. The lateral tubercles are usually well-defined (Figure 1e) and the transverse carinae are always present, joining the sides of the base of the central tubercle. The main differences then between the *marginatus* and *falsus* types are the ‘free’ central tubercle and more distinct lateral tubercles in the latter, with other differences involving connections between other structures.

*The MFS type of* Tonantzin tepetl *n. sp. and how it relates to the other types*. Our report herein of the new genus *Tonantzin* documents a new and distinct evolutionary lineage within Proculini (Figure 2) also defined in part by a novel configuration of mesofrontal structure characters. We argue that the MFS in *Tonantzin* represents a fifth “type” (*tepetl*), distinct in several critical ways from the four original types defined by Reyes-Castillo [5]. As we describe above, in this new type the central tubercle is very large, elongate and straight (not bent in lateral view), is free (surpassing the frontal margin) and projects anteriorly (Figure 3 and Figure 4). The lateral tubercles are perhaps of most significance in defining the *tepetl* type. Unique for Proculini, the lateral tubercles are large and distinct, elongate, narrow and oriented anteriorly (subobliquely). Critically, however, they are also distinctly separated from and positioned farther back from the base of the central tubercle (Figure 3 and Figure 4), and also raised slightly above it (they are at a slightly higher plane). Both the inner and outer sides of the tubercles are also well marked and without a transverse carinate connection with the central tubercle, rendering these cuticular protuberances distinct from the central tubercle. Also distinctive to the *tepetl* type, however, are the low convex ridges arcuately extending from the base of the central tubercle, nearly reaching the supra orbital ridge (Figure 3 and Figure 4). These “frontal” ridges may represent a novel MFS character as they are continuous with the central tubercle but also cannot easily be homologized with the frontal ridges given the significant difference in orientation/position and fine structure. However, the lateral termination of these ridges somewhat resembles the internal tubercles often present at the ends of the more medially positioned frontal ridges in other taxa (these also then forming a distinct angulate V), which suggests more work is needed to ascertain the homology of this structure. Similar ridges can be observed in species with an MFS of the *striatopunctatus* type (e.g., *Odontotaenius zodiacus* [Truqui, 1857], *Y. cameliae*). However, it is not possible to discern if “frontal” ridges of the *tepetl* type and those observed in species with the *striatopuncatatus* type are homologous. They both appear to be a continuation of the base of the central tubercle, rather than distinct ridges.

The *striatopunctatus* and *bicornis* types are therefore simple MFS types in terms of the number of constituent structures that are usually present. But for our discussion they are also the most qualitatively different from each other and in a significant conceptual way: in the absence of the other MFS types there would be no basis on which to consider the lateral tubercles of the *bicornis* type and central tubercle of the *striatopunctatus* type as having a structural relation to each other, i.e., as forming independent substructures of a larger complex structure. It is only by comparison with species showing apparently both of these structures that it is feasible to hypothesize that they are homologues with similarly positioned structures in passalids that have both structures, and therefore related as parts of the mesofrontal structure. In the *Tonantzin* type, the unique positioning of the lateral tubercles in relation to the central tubercle also calls into question the criteria on which these structures can be judged as homologous to the transverse tubercles of species showing the *marginatus* and *falsus* types (with which the *tepetl* type is most similar) and, most importantly, to those of the *bicornis* type. 

*How to define the boundaries of MSF?* A fundamental problem in applying MFS types to the classification of Proculini (and Passalidae) consists of determining the boundaries of the MFS. If this cannot be achieved objectively then the shift from using MFS types to a more atomistic approach may seem justified. For example, it is unclear whether characters such as the frontal ridges, internal and external tubercles and other structures unique to certain taxa should be considered part of the mesofrontal structure. It is also unclear how modifications of certain structures affect their interpretation, as in the cases where transverse carinae are present but distinct lateral tubercles are seemingly absent (e.g., *Yumtaax laticornis* [Truqui, 1857]) or when the frontal ridges terminate in a tubercle at the frontal margin (e.g., *Vindex* Kaup, 1871, *Pseudacanthus*), which suggests these tubercles may be homologous to the internal tubercles usually present on the frontal region but distinctly posterior to the frontal margin. According to Reyes-Castillo [5], the MFS consists only of the central and transverse carinae/lateral tubercles (in the broad sense). However, Boucher [1] expanded the delineation to include other tubercles, sulci and fossae surrounding these structures. 

We examined cleared specimens from a range of taxa in order to determine if any structural features inside the head could be used to help define the boundaries of the MFS and to provide a preliminary assessment of possible criteria that may be used to delimit the MFS to include only the central tubercle, lateral tubercles (and transverse carinae) and frontal ridges (up to and including only the internal tubercles). The following points arise from our examination of cleared specimens: 

1. In Proculini we have examined, the internal structures useful for providing landmarks by which to delimit the MFS include the epicranial sutures, oblique tentorial arms, tentorial pits (where the tentorial arms join the cranium) and the internal position of the membranous clypeus. The epicranial sutures in Proculini consistently terminate posteriorly in a sclerotized internal structure (antennal fossae Boucher [1]). From the junction of the epicranial suture and this structure oblique tentorial arms extend to and join the tentorial arms at the point where the tentorial arms connect with the tentorial pits. Extending between the ends of the epicranial sutures (at their junction with the oblique tentorial arms) there is a weak transverse internal ridge delimiting the posterior attachment of a membranous flap (probably the clypeus). Only the central tubercle and lateral tubercles (including the transverse carinae) are consistently contained within these borders (oblique arms, transverse internal ridge, pits) and thus it may be reasonable to consider these structures as landmarks denoting positional criteria for defining what structures are part of the MFS considering the diverse and variously interconnected carinae, tubercles and sulci on the dorsal side of the head. The frontal ridges are not always contained within this boundary but often extend anteriorly from the central tubercle to the frontal margin.

2. Our observations indicate that the central tubercle and usually the lateral tubercles and transverse carinae have both internal and external structure: when well-developed it is obvious that they represent substantial evaginations of the cuticle (rather than primarily cuticular thickening) thus expanding the cranial volume. On the other hand, other cuticular structures of the head such as the frontal ridges, internal and external tubercles are seemingly largely external structures (at least when only weakly developed), without substantial or any cuticular evagination.

3. The position of the tentorial pits (the dorsal connection of the tentorial arms) in relation to the lateral tubercles may be used as a landmark to identify these tubercles and allow justification of primary homology assessment across different taxa. The tentorial pits are consistently posterior to the base of the tubercles even though the distance between tubercle and pit can vary (in some taxa the pits are coincident with the base of the tubercle, e.g., *Spurius bicornis* [Truqui, 1857]). So far as we know these tubercles are never positioned posterior to or lateral to these pits. This positional relation may, therefore, be used to justify considering tubercles in this position as homologues across taxa and suggests an independent criterion for considering the paired posterior tubercles of the *bicornis* type as part of the MFS.

The idea that internal structures may be used to help define the mesofrontal structure is not arbitrary since the position of these structures in relation to muscle mass and attachment sites and other organs may relate to the functional significance of the MFS. These structures combined with the occipital sulcus and frontal fossae may serve to provide a unified basis for delimiting the mesofrontal structure. Future work could involve thin sectioning of the head of exemplars from each type in order to identify internal tissues in the vicinity of the MFS (e.g., muscle attachment sites within the MFS).

*Broader applicability of the mesofrontal structure in Passalidae.* The system of MFS types developed by Reyes-Castillo [5] works well for the generic-level classification of Proculini such that congeners have consistent MFS types and subtypes. However, elsewhere in Passalinae (e.g., Passalini), these types do not seem to apply, nor is there consistency within genera to the same extent as there is in Proculini e.g., [33,34]. This might seem to limit the usefulness of the composite “type”-based classification of this complex structure in the broader context of Passalidae systematics. Two points might be the main reasons why the application of the composite “type” system is difficult to apply outside Proculini: (1) Incorrect circumscription of taxonomic groups; and (2) the existence of different MFS configurations (“types”) outside of Proculini. 

In the case of Passalini, this lack of utility (one-to-one correspondence of MFS types and genera within the tribe) could be misleading and might instead be due to the lack of taxonomic organization in the tribe. When Fonseca & Reyes-Castillo [35] described *Passipassalus* Fonseca & Reyes-Castillo, 1993, they indicated that the *striatopunctatus* type of MFS was one of the distinctive characters of the genus. This character was maintained in the re-description of the genus by Jiménez-Ferbans & Reyes-Castillo [36]. These authors conducted a phylogenetic analysis for *Paxillus* MacLeay, 1819 and *Passipassalus*, and some characters derived from the MFS were recovered as synapomorphies for these genera. Thus, according to these authors, the absence of lateral tubercles and the very free apex of the central tubercle are synapomorphies that help delimit *Passipassalus*, while all species of *Paxillus* have an MFS that approximates the *marginatus* type. Mesofrontal structures of the *marginatus* type, or something close to it, are the most common in the Passalini genera *Spasalus* Kaup, 1869, *Ptichopus* Kaup, 1869 and *Ameripassalus* Jiménez-Ferbans & Reyes-Castillo, 2014. 

In contrast, the correspondence between MFS types and taxonomic groupings does not apply in the genus *Passalus* F., 1972 (which includes 70% of the species of Passalini). Nonetheless, it is important to mention that *Passalus* has been recovered as paraphyletic or polyphyletic in almost all phylogenetic analyses of Passalidae carried out so far [1,34,36,37], as have the recognized groups (subgenera and species groups) within the genus. Thus, the lack of correspondence between the MFS types and the *Passalus* groupings may be due to the incorrect delimitation of groups within this genus.

Also, application of the Reyes-Castillo [5] type system can be difficult outside and sometimes inside of Proculini, because the MFS types of the taxonomic groups do not conform to those described by Reyes-Castillo [5]. For instance, in some Passalini species with a MFS that approximates the *marginatus* type, the configuration is not the same. In the *marginatus* type, the central tubercle is fused to the frontal ridges by a single longitudinal carina or pair of carinae. In Passalini species with an MFS similar to the *marginatus* type, the frontal ridges are fused directly and raised to the same plane as the central tubercle, causing a slope that expands laterally. Additionally, the lateral tubercles are sometimes indistinguishable from those slopes. Thus, if the use of MFS types is to be applied outside Proculini, it might be useful to characterize new configurations of this structure. 

### 4.2. Biogeography and Micro-Endemism

The altitudinal stratification of passalid diversity is well-documented (e.g., [4,38,39]), resulting in different degrees of endemism within the family. The genus *Heliscus* Zang, 1906 is a clear example of this phenomenon in the mountains of Mexico. For the Mexican species of *Heliscus*, generally, the more restricted their altitudinal distributions are (and, more importantly, the higher the lowest elevation of their distribution is), the more restricted their latitudinal distributions become [13]. For example, the altitudinal range of *Heliscus tropicus* (Percheron, 1835) is 300–2300 masl, and this species is restricted to the Sierra Madre Oriental, Sierra de los Tuxtlas, Sierra de Juárez, and the Northern Mountains in Chiapas. In contrast, the altitudinal range of *Heliscus vazquezae* Reyes-Castillo & Castillo, 1986 is 900–1300 masl, and the species is restricted to the Sierra Madre Oriental [13]. Some montane species in Mexico also show an extreme degree of local endemism. For instance, *Y. laticornis* is only known from Pico de Orizaba, Veracruz, and *Y. cameliae* is known only from a small patch of oak forest in Puerto del Aire, Acutzingo, Veracruz [9]. Both of these species are known from altitudes above 2000 masl. *Tonantzin tepetl* is only known from Cofre del Perote, Veracruz, at altitudes above 2800 masl. 

The Cofre de Perote is an extinct volcano, part of the Las Cumbres Volcanic Complex (Puebla, and Veracruz) that comprises Quaternary-aged volcanoes [40]. Along with The Pico de Orizaba (5747 m) the Cofre de Perote (4282 m) are the two highest montane elevations of the country [41]. In the vegetation gradient of Cofre de Perote you can find tropical and temperate forests and montane grasslands [42]. The vegetation between 2000 and 3000 masl is characterized by pine-oak forest, and above the pine-oak forest grows fir (*Abies*) forest [43]. *Tonantzin tepetl* is known from the transition of these two ecosystems.

The sampling of Passalidae from lower altitudes has been much greater than that at higher elevations in the area of Cofre del Perote, so the occurrence of *T. tepetl* suggests that the species is a montane endemic. Arguably, we can consider *T. tepetl* a local endemic of Cofre de Perote. Mexico is a country with one of the most extensively sampled passalid faunas [1]. However, exemplars of *Tonantzin tepetl* are only known from this restricted area. This suggests that other populations of this species, or potentially related species, might be found in other high-elevation ecotones between pine-oak and fir forests elsewhere in the region.

## 5. Conclusions

At the generic level, the tribe Proculini is most diverse in Mesoamerica, and in Mexico the majority of passalid species are Proculini. Even though the passalid fauna of Mexico is very well documented, new taxa are constantly being described. The description of new taxa is mainly associated with the discovery of montane endemics or new circumscriptions of previously known taxa. After close examination of the specimens of *Tonantzin tepetl* and considering “The General Lineage Concept” [44,45] this new potentially narrow endemic genus can now be understood in light of Mexican passalid biodiversity. 

From the standpoint of comparative morphology and primary homology assessment, for example, as discussed by Wilkinson [46], we find value in considering the MFS a complex character composed of several constituent characters and suggest that the evolutionary significance of this structure is better captured by viewing it this way: the five mesofrontal structure types may represent alternative endpoints along a trajectory of morphological change. 

## Figures and Tables

**Figure 1 insects-10-00188-f001:**
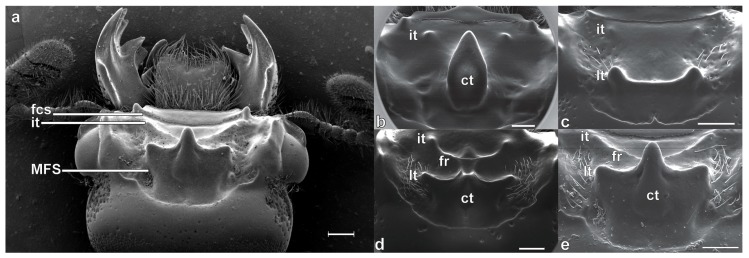
Scanning electron micrographs of mesofrontal structures of selected Passalidae, dorsal view. (**a**) Head capsule of *Yumtaax recticornis* (Burmeister, 1847), for context. (**b**) The “*striatopunctatus*” type; *Odontotaenius striatopunctatus* (Percheron, 1835). (**c**) The “*bicornis*” type; *Spurius bicornis* (Truqui, 1857). (**d**) The “*marginatus*” type; *Popilius* sp. (**e**) The “*falsus*” type; *Y. recticornis*, detail of (**a**). Abbreviation list, fcs = frontoclypeal suture, it = internal tubercles, MFS = mesofrontal structure [5], ct = central tubercle, lt = lateral tubercles [5], latero-posterior tubercles [1], fr = frontal ridges [5], postero-frontal ridges [1].

**Figure 2 insects-10-00188-f002:**
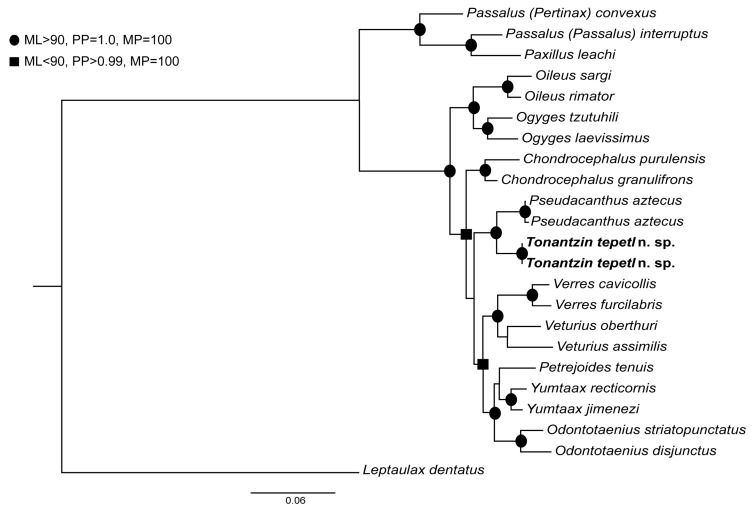
Maximum likelihood tree resulting from the partitioned analysis. Nodal support is indicated for relationships with strong statistical support.

**Figure 3 insects-10-00188-f003:**
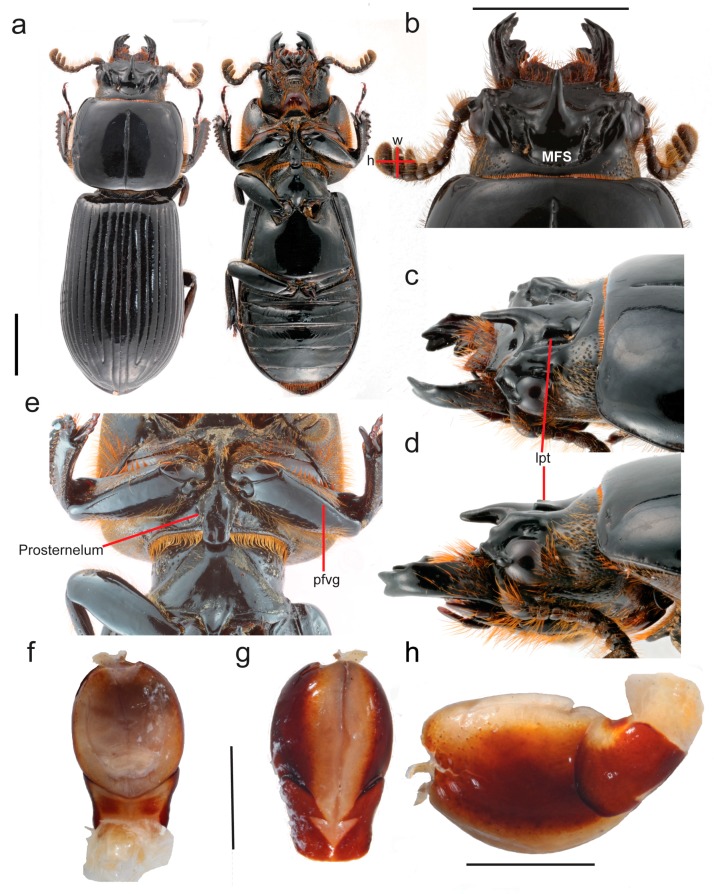
*Tonantzin tepetl* n. sp. (**a**) Habitus, dorsal and ventral views. Head views: (**b**) dorsal; (**c**) oblique; (**d**) lateral. (**e**) Prothorax, ventral. Aedeagus: (**f**) dorsal; (**g**) ventral; (**h**) lateral. Scale bars: 5 mm (3a–3d); 1.5 mm (3f–3h). Labels: MFS = mesofrontal structure, lpt = latero-posterior tubercles, pfvg = profemur ventral grove. a–e (allotype); f–h (paratype).

**Figure 4 insects-10-00188-f004:**
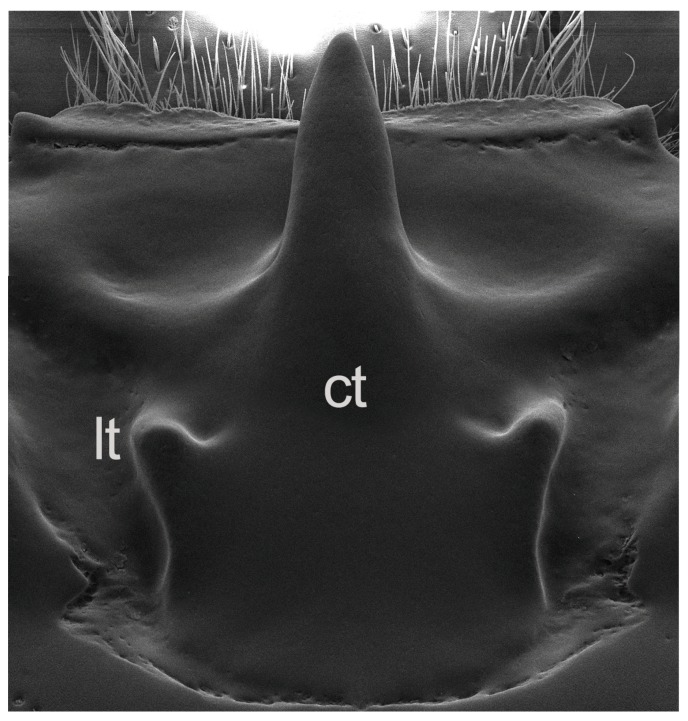
Scanning electron micrograph of the MFS type *tepetl*. Acronym list, ct = central tubercle, lt = lateral tubercles [5], later-posterior tubercles [1].

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
