# Peer review of "Tonantzin, a New Genus of Bess Beetle (Coleoptera, Passalidae) from a Montane Subtropical Forest in Central Mexico, with a Review of the Taxonomic Significance of the Mesofrontal Structure in Proculini"

_insects, 2019, doi:10.3390/insects10070188_

Round 1
Reviewer 1 Report
The manuscript reports on a new species of Passalidae from Mexico and describes its characters and placement well. However, I am not quite convinced that the description of a new genus is really warranted. Possibly, it could be included in the genus "Pseudacanthus" to which it is closely related. Even if not all morphological characters agree – it is not unusual that morphological concepts of genera need to be adjusted. If the types of the MFS are considered so important and distinct I would suggest that they should be traced on the phylogeny to visualize their evolution and convey the concepts of distinct character gaps to the reader. Also, I am curious why cox1 sequences have not been used for the phylogeny reconstruction.
Below a few minor issues that should be fixed:
Abstract: Lines 15-25: there are changes from present tense to past tense. I would stick with the former, such as "We define..." "We propose..."
Line 32: no need for insertion in parentheses. Try to rephrase in one or two sentences.
Lines 42, 50 and later: Is the term "circumscriptions" really adequate? I would either speak of "descriptions" or "delineation" of genera, or possibly the "concept of a genus". Circumscription may not be wrong, but it is used very often – maybe think of better alternatives.
Lines 184 ff: The use of the nuclear gene CAD is a good thing, but I am wondering why cox1 was not included. Usually, this is sequenced at first, also for "barcoding" (identification) purpose. So, was it sequenced? If yes, did it support a different phylogeny?
Line 224: the genus Tonantzin name should be in italics and bold format;
I think the usual term would be "by present designation" instead of "designated herein"
Line 226: Since the name is a bit unusual I feel it should be stated explicitly what gender it is. If new species will be added in future and their names are adjectives or participles, it needs to be clear which gender to follow.
Line 228 and throughout the text: "Tonantzin" and "Tonantzin tepetl" should be in italics
Line 230: Is it really necessary to state "Adult" ? In descriptions of beetles usually the adult is described (which is also evident form the illustrations) and in case the larva is described, this is mentioned specifically.
Line 262: "Type Material" should be Type material". I would delete "Three specimens (two male(s?), one female" since this is redundant.
Line 267: The "allotype" is actually only a special paratype. In fact, there is only a holotype and non-holotypes, i.e. paratypes. So, I would continue with the Heading "Paratypes" and then indicate for one of them "designated as allotype". The collection codon could be provided simply in parentheses, without "deposited in"
Line 280: I feel "Length should come first and it should be separated by a period. Then continue "Body black, shiny.
Line 303 and following: I think it is not necessary to end every subheading with a period. Why not say "Elytra shiny, anterior border rectangular.
Wings fully developed."
Line 307: see above: I would suggest: Abdomen with visible tergites glabrous;
Lines 324 ff: If the generic concept of Petrejoides is unclear and polyphyletic, then it is no wonder that the new genus cannot be placed into it with confidence.
Author Response
Comments from reviewer 1 are addressed below. We accepted all grammatical and word choice suggestions from Reviewer 1 with the following exceptions:
Lines 42, 50 and later:
Reviewer: Is the term "circumscriptions" really adequate? I would either speak of "descriptions" or "delineation" of genera, or possibly the "concept of a genus". Circumscription may not be wrong, but it is used very often – maybe think of better alternatives.
Answer: We changed the word circumscription when we felt it was warranted. We kept the word circumscription in cases when boundaries for groupings were defined. In the authors opinion the word is adequate. In our view, classes of inanimate objects that are constant and not changing in time, are "defined" by sets of characteristics. Conversely, species are parts of lineages that change over time and are "circumscribed" (e.g., see "Species Concepts and Phylogenetic Theory" by Wheeler and Meier 2000).
Line 303 and following, and Line 307:
Reviewer: I think it is not necessary to end every subheading with a period. Why not say "Elytra shiny, anterior border rectangular.
Wings fully developed."
Reviewer: see above: I would suggest: Abdomen with visible tergites glabrous;
Answer: We think the subheading with a period is necessary to keep consistency throughout the description. There are useful characters described, although not many within these subheadings. Thus, the length under the subheadings, make them seem unnecessary. But we decided to keep them an maintain consistency.
Comments.
Reviewer: manuscript reports on a new species of Passalidae from Mexico and describes its characters and placement well. However, I am not quite convinced that the description of a new genus is really warranted. Possibly, it could be included in the genus "Pseudacanthus" to which it is closely related. Even if not all morphological characters agree – it is not unusual that morphological concepts of genera need to be adjusted.
Answer: Two options were considered by the authors before the decision of ranking the species in a newly described genus. Within those options we considered the placement of T. tepetl in Petrejoides but the phylogenetic evidence, and the problems within the circumscription of Petrejoides did not supported this decision. A second option, based on the phylogenetic evidence, was to placed T. tepetl within Pseudacanthus. However, to include this new species in Pseudacanthus we would have to completely change the delimitation of this genus, rendering Pseudacanthus undefinable. For example, the diagnostic characters of Pseudacanthus is the position of the internal tubercles (which expand beyond the anterior border of the clypeo-frons, interrupting the frontoclypeal suture); conversely, Tonantzin lacks the internal tubercles at all. Additionally, other minor characters of the Tonantzin do not fit the definition of Pseudacanthus. Furthermore, the larval morphology of Pseudacanthus suggest this genus might not be monophyletic, also making the placement of T. tepetl within this genus a risky choice. In addition, expanding the concept of Pseudacanthus would signify the re-circumscription of genera that are morphologically close to Pseudacanthus (e.g., Vindex, Proculejus, Xylopassaloides); thus, creating instability in the genera of the tribe. Based on this evidence, we consider the species required the description of a new genus, and we found no evidence of the contrary. Furthermore, Tonantzin is a taxonomic hypothesis proposed by the authors, and as such it could be subjected to additional testing and rejected should the evidence support this decision.
Reviewer: If the types of the MFS are considered so important and distinct I would suggest that they should be traced on the phylogeny to visualize their evolution and convey the concepts of distinct character gaps to the reader.
Answer: The authors agree that tracing the MFS in the phylogeny is an important source of information. However, we need a much greater taxa sampling to make this of any significance. The phylogeny on this study was not designed to test the MFS under a phylogenetic context, but rather test the position of Tonantzin relative to other taxa. Although a very valuable suggestion, we consider that tracing of the MFS in this particular phylogeny would give room to misinterpretation, given the taxa sampling and context of the study.
Lines 184 ff:
Reviewer: The use of the nuclear gene CAD is a good thing, but I am wondering why cox1 was not included. Usually, this is sequenced at first, also for "barcoding" (identification) purpose. So, was it sequenced? If yes, did it support a different phylogeny?
Answer: We did not amplified CO1 for this study. We agree that CO1 gene is used for DNA barcoding. However, the species delimitation in this study did not include DNA data. The purpose of the phylogeny was to ensure Tonantzin did not belong to any other genera of Proculini. Although, CO1 (along with 12s, and 28s) has been successfully used to do species delimitation in Passalidae (e. g. Beza-Beza et al. 2017), the data did not provide good resolution when it came to resolve generic level relationships. Considering the poor performance of CO1 in generic level relationships, we decided to use CAD and 28s which performed better at this task.
Line 226:
Reviewer: Since the name is a bit unusual I feel it should be stated explicitly what gender it is. If new species will be added in future and their names are adjectives or participles, it needs to be clear which gender to follow.
Answer: We have assigned the genus name feminine gender, in compliance with IZCN articles 30.2.2, and checked the species name comply with gender agreement in compliance with article 31.2.
Lines 324 ff:
Reviewer: If the generic concept of Petrejoides is unclear and polyphyletic, then it is no wonder that the new genus cannot be placed into it with confidence.
Answer: We agree with the reviewer on this statement. Additionally, the phylogenetic data presented does not support the inclusion of Tonantzin tepetl in Petrejoides.
Reviewer 2 Report
The manuscript is interesting, and surely I can suggest the pubblication on Insects. Here below you’ll find some comments:
rows 33-34. Since the Mexico is partly included in the Nearctic Region, the phrases should be re-written to account for the exact position of the collection locality of the new Passalid species
row 74. The meaning of “individual structures” is unclear, are the differences related to species identification?
row 121. Do you mean“pinned”?
rows 157-183. Perhaps also the COI sequence should be included, since it is the reference for species identification (i.e., DNA Barcoding)
row 234. “Structure” is in capitals
Author Response
We accepted all grammatical and word choice suggestions from Reviewer 2. Comments from the reviewer are address below.
Lines 33-34:
Reviewer: Since the Mexico is partly included in the Nearctic Region, the phrases should be re-written to account for the exact position of the collection locality of the new Passalid species
Answer: we added “in Mexico most of the diversity of Passalidae is found in the subtropical region.” To the manuscript to clarify the species richness distribution in Mexico
Line 74:
Reviewer: The meaning of “individual structures” is unclear, are the differences related to species identification?
Answer: we changed the phrase to read “The presence or absence of the individual characters which comprise the MFS sensu Boucher [1] was important in his species delimitations”
Line 157-183
Reviewer: Perhaps also the COI sequence should be included, since it is the reference for species identification (i.e., DNA Barcoding)
Answer: We did not amplified CO1 for this study. We agree that CO1 gene is used for DNA barcoding. However, the species delimitation in this study did not include DNA data. The purpose of the phylogeny was to ensure Tonantzin did not belong to any other genera of Proculini. Although, CO1 (along with 12s, and 28s) has been successfully used to do species delimitation in Passalidae (e. g. Beza-Beza et al. 2017), the data did not provide good resolution when it came to resolve generic level relationships. Considering the poor performance of CO1 in generic level relationships, we decided to use CAD and 28s which performed better at this task.